# Aspects of Breeding Performance of Scopoli's Shearwater (*Calonectris diomedea*): The Case of the Largest Colony in Greece

**Georgios Karris** [1,*], **Stavros Xirouchakis** [2], **Konstantinos Poirazidis** [1], **Marios-Dimitrios Voulgaris** [3], **Anastasia Tsouroupi** [1], **Spyros Sfenthourakis** [4] and **Sinos Giokas** [5]

1 Department of Environment, Ionian University, 29100 Zakynthos, Greece; kpoiraz@ionio.gr (K.P.); e16tsou@ionio.gr (A.T.)
2 Natural History Museum of Crete, School of Science & Engineering, University of Crete, P.O. Box 2208, 71409 Crete, Greece; sxirouch@nhmc.uoc.gr
3 ENVIR-Environmental Research Services, 84011 Folegandros, Greece; m.d.voulgaris@gmail.com
4 Department of Biological Sciences, University of Cyprus, P.O. Box 20537, 1678 Nicosia, Cyprus; sfendourakis.spyros@ucy.ac.cy
5 Section of Animal Biology, Department of Biology, University of Patras, 26500 Patra, Greece; sinosg@upatras.gr
* Correspondence: gkarris@ionio.gr

**Abstract:** Here we report, for the first time, aspects of the breeding performance of Scopoli's Shearwater (*Calonectris diomedea*) in one of its largest colonies in Europe, i.e., in the Strofades island group. We describe the chronology of the main events in the breeding cycle of this species on Stamfani Island, the largest island of this island group, including the evaluation of breeding performance and the influence of ecological factors (nesting habitat, nest type, nest-entrance orientation, and occupation rate per nest) on breeding success. The Scopoli's Shearwater colony of Stamfani Island exhibited a high degree of breeding synchrony and nest site tenacity. The data obtained by monitoring 472 nests during five consecutive years (2008–2012), showed a breeding success (fledging per occupied nest per year) of up to 66.6%. In addition, hatching success (chick hatched successfully per egg laid) was 76.9%, and fledging success (fledging young per chick hatched successfully) was 86.8%. We also observed annual variations in breeding performance during that period. These results seemed to be influenced positively by the breeding experience of the pair. Furthermore, the type of nest site and the nest-entrance orientation were found to have an effect on breeding success rates, whereas the nesting habitat did not, indicating low predation risk by rats.

**Keywords:** seabird conservation; marine ecosystem; population dynamics; Procellariiformes; Strofades; National Marine Park of Zakynthos



## 1. Introduction

Seabirds are a diverse group of more than 400 species, spending part or all of their lives interacting with oceans, e.g., by foraging and migrating over them [1]. They constitute one of the most threatened groups of birds [2,3], facing ecological challenges such as invasive alien species, incidental (by-catch) mortality on fishery gears, and climate change/severe weather [4]. These marine top predators are, in general, long-lived birds with delayed maturity and low annual reproductive rates. Many seabird species have long lifespans (>30 years) and high adult survival, with less than 10% dying each year, and most of them show delayed adult maturity, commencing breeding after the age of three years [5]. They also show low fecundity (1–3 eggs clutch per breeding attempt) and an extended chick-rearing period. In addition, the fact that they are sensitive to variations in food supply, readily visible at sea, and dependent on land for breeding allows for a better understanding of their population trends and the assessment of possible threats to their conservation [4]. Consequently, they are recognized as important bio-indicators of marine ecosystems that are useful in evaluating the environmental disturbance of marine biotas [6–8].

The order Procellariiformes constitutes a group of K-selected pelagic seabirds that exhibit high plasticity to the marine environment, covering large distances and living over the sea outside the breeding season [1,9]. They form colonies and nest in burrows, crevices, and cavities under boulders on isolated, inaccessible, and mainly uninhabited islands and islets while they lay one egg per breeding pair as a result of their adaptation to highly dynamic marine ecosystems [9]. The balance or imbalance of energy intake and demand determines the fitness of individuals and may negatively affect their reproductive success to the point where they may temporarily interrupt breeding (sabbatical year) [10]. That phenomenon is not so uncommon among birds such as shearwaters, especially if we consider inexperienced individuals [11], the individuals' time of arrival from their wintering areas to their colonies [12], as well as their colonies' characteristics [13].

The burrowing seabird targeted in this study is a long-lived migrant procellariid species, well-known for nest site tenacity, mate fidelity, pelagic distribution even during breeding, and nocturnal behavior as an adaptative strategy to avoid terrestrial predators [14–17]. The breeding sites of the Scopoli's Shearwater (*Calonectris diomedea*), which lay only one egg per pair, are widespread across the Mediterranean Basin, whereas its wintering grounds are mainly located in the North Atlantic Ocean where the Canary Current occurs and, secondly, in the pelagic and coastal equatorial areas of the Eastern Atlantic, as well as in the South Atlantic Ocean along the west coast of Southern Africa [18]. Scopoli's Shearwaters have strong interactions with fishery operations in the Ionian Sea. These can be negative, such as incidental mortality (by-catch) caused by various fishery gears like bottom longlines, surface longlines, and gillnets [19], or positive, since trawlers provide a significant amount of discarded demersal fish extensively exploited by this scavenger, as highlighted in a recent study [20].

The well-documented knowledge of breeding biology and nesting habitat use constitute crucial information for seabird species that we aim to conserve or plan to use as bioindicators of marine ecosystems [1]. The Strofades Islands in the Ionian Sea are considered one of the three Scopoli's Shearwater European strongholds which host colonies of more than one thousand breeding pairs each [21]. Despite this fact, no studies on the breeding rates of Scopoli's Shearwater in the Strofades colony have been conducted so far. To address this lack of knowledge, we examined the breeding performance of this procellariiform species in the Strofades Islands for the first time, for five consecutive breeding seasons. More specifically, we assessed the relationship between breeding success and (a) type of nest sites, (b) nest entrance and orientation features, (c) nesting habitat reflecting different levels of rat presence and possible predation, (d) rate of nest site occupation throughout sampling years and its correlation with breeders' experience, and (e) dimensions of eggs. The results of our research can be used to define conservation management priorities by providing reliable information about the Scopoli's Shearwater colony and its population status within the Strofades isolated insular area. Several studies carried out on different Scopoli's Shearwater colonies in the Mediterranean provide additional emphasis on the benefits of a systematic monitoring of the species' reproductive effort and success rates [22–25].

## 2. Materials and Methods

### 2.1. Study Area

The Strofades island complex (37°15 N, 21°00 E) constitutes a remote group of two small and flat islets (the highest point is 22 m.a.s.l), namely Stamfani and Arpyia, surrounded by several rocks. The total insular area is about 4 km$^2$, lying approximately 32 n.m. to the south of Zakynthos Island and 26 n.m. to the west of the Peloponnese peninsula, within the National Marine Park of Zakynthos in the Ionian Sea. These islets show a variety of habitats, such as rocky coastline, broadleaved forest, brushwood, agricultural fields, including cereal cultivations, reedbeds, and sandy beaches, and a *Juniperus turbinata* Gussone (syn. *Juniperus phoenicea* Linnaeus) forest, which constitutes one of the few remaining high forest formations of this species in the Mediterranean [26] (Figure 1). The Strofades

islands host the largest colony of Scopoli's Shearwater in Greece, with ca. 5550 breeding pairs [27], as well as one of the most significant stopover sites in the eastern Mediterranean for passerines that migrate from Africa to the Palearctic during spring [28]. They are also characterized as a multi-invaded insular ecosystem where rodents and cats play the roles of mesopredators and superpredators of shearwaters, respectively. More specifically, the black rat (*Rattus rattus*) seems to be the most abundant predator on Stamfani Island where, in 2011, rat density was estimated at 5.37 individuals/ha [29].

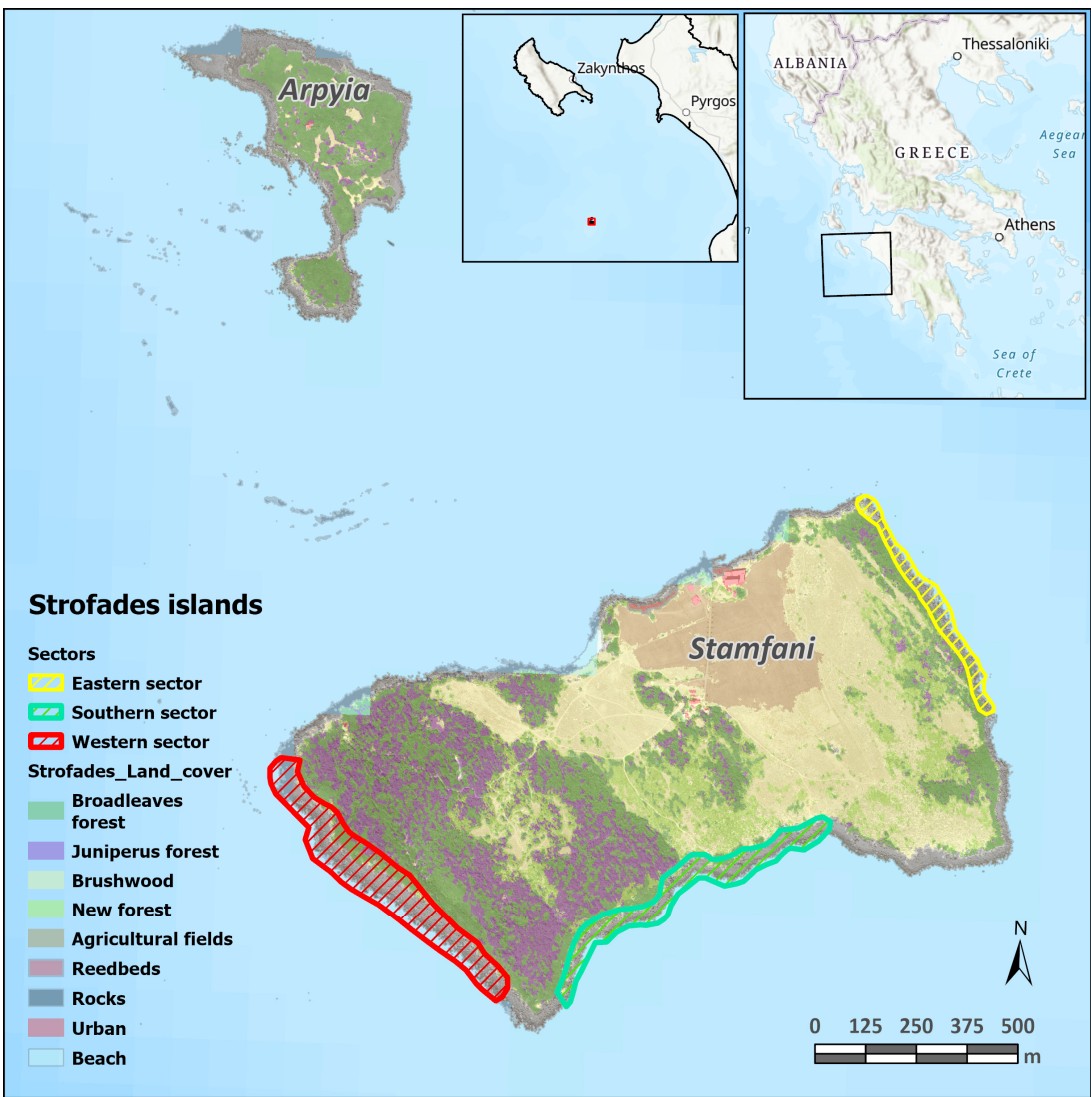

**Figure 1.** Location of Strofades islands and the habitat types found on them. Sectors of the Stamfani coastline where breeding performance of Scopoli's Shearwater was monitored during 2008–2012 are also shown.

*2.2. Breeding Performance*

The Scopoli's Shearwater breeding performance was monitored on Stamfani Island for five consecutive years (2008–2012). The methodological process that was followed ensured an accurate recording of the birds' normal behavior by minimizing the impacts of possible disturbances caused during fieldwork [30]. The field techniques applied were in accordance with the standards of previous studies [31–34]. In addition, we took into consideration the outcome of a pilot study on Stamfani Island during the breeding season of 2007, aiming to identify a suitable coastline for nesting, where more than 95% of the nests were found within a 20 m wide area only 5 m from the coastline [27].

During the 2008–2012 period, a research team of 3–4 members checked (during moonless nights) three sectors of coastline with different nest site quality, using stratified sampling so as to combine the simplicity of random sampling with the potential increase in survey reliability [35]. More specifically, the western sector was characterized by a rocky coastline with sparse shrub cover and bare soil suitable for excavation, the southern sector was characterized by a rocky coastline with dense maquis vegetation cover and very little bare soil, whereas the eastern sector was the least accessible, characterized by a harsh rocky coastline without shrub cover and bare soil (Figure 1). Each sector was treated as a different nesting habitat reflecting different levels of rat presence because of the landscape, combined also with observations of rodent droppings at different abundances per sector [29]. Thus, the classification of potential predation pressure at each sector showed three levels of actual rat presence, namely, a high level in the western sector, a medium level in the southern, and a low level in the eastern.

From 25 May to 15 October of each breeding season, nest sites distributed throughout all sectors were systematically monitored by using a burrowscope with a wide-angle CCD camera and 2 m cable, so as to ensure the minimum bias in evaluating breeding performance. Three field visits were made each year to monitor the Scopoli's Shearwater reproductive effort. The first visit took place between 25 May and 10 June and aimed to check potentially occupied nests according to the pilot study of 2007 and to mark active nests (those with a breeder incubating). The second took place in the second fortnight of July, collecting information on the breeding effort at the hatching stage, while the third took place during the last 10 days of September and the second fortnight of October so as to check for fledglings (Figure 2). It was assumed that no further mortality incidents occurred after our last visit every breeding season. According to the obtained data, the breeding success (% fledglings per egg laid), hatching success (% chicks hatching successfully per egg laid), and fledging success (% fledglings per egg laid) were estimated. Data on breeding performance were recorded in datasheets along with general information (e.g., date, time, evidence of predation, etc.), sector name, and GPS waypoints defining the exact location of each nest site and nesting habitat type. Birds (breeders, but mainly fledglings) were also ringed, blood sampled, and measured. The collected data were entered into a database which was interactively connected with GIS software (ArcGIS Pro 3.0) so as to be depicted in digital maps with respective geographical parameters.

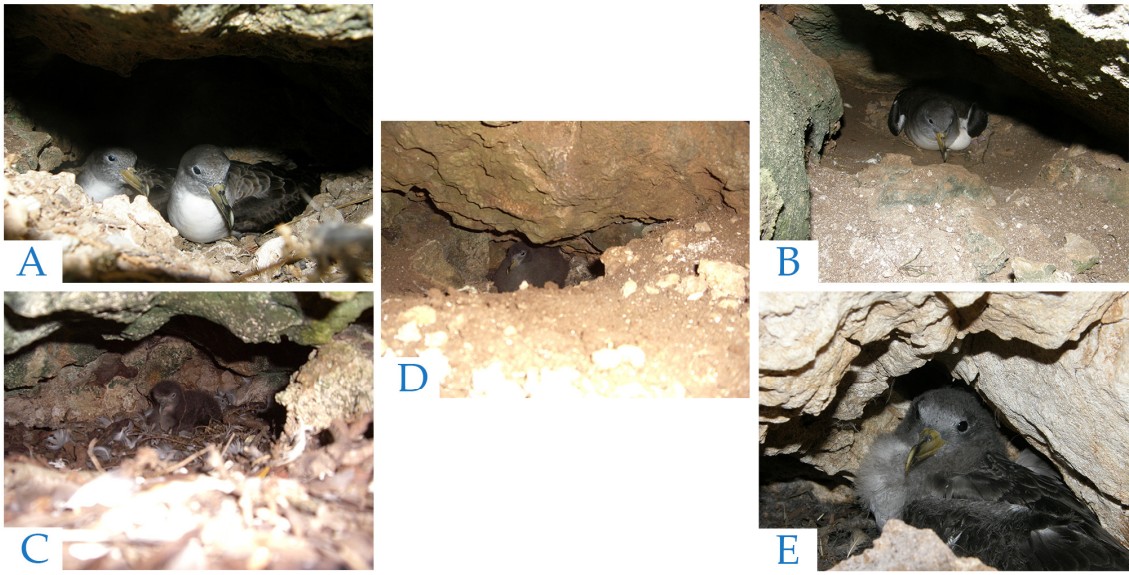

**Figure 2.** Different stages of Scopoli's Shearwater breeding performance in Stamfani colony: (**A**) shearwaters mating, (**B**) egg-laying, (**C**) nestling of a few days old after egg hatching, (**D**) chick of a few weeks old, (**E**) fledgling.

### 2.3. Factors Influencing Breeding Success

Several environmental factors were tested for their possible impacts on the breeding success of Scopoli's Shearwater, namely, the nesting habitat features reflecting different levels of rat presence/predation pressure, as already mentioned, the type of nest site, the orientation of the main entrance of each nest, the rate of nest site occupation throughout sampling years reflecting different level of breeders' experience, and eggs' dimensions. More specifically, the nest sites of Scopoli's Shearwater in the Stamfani colony were categorized into five different types: natural deep cavities under cliff cover (cliff cover), natural shallow cavities under stones (stone cover), cavities under shrub cover (shrub cover), rock cavities among fallen boulders (crevices), and burrows excavated by breeders (burrows). The orientation of the main entrance of each nest site was categorized as western (west), eastern (east), northern (north), or southern (south). Nest sites were also ranked from 0–5 given the total number of successful breeding occupation events during the 2008–2012 sampling period, assuming that nests with higher scores are occupied by more experienced breeders since the species is characterized by a high degree of nest site tenacity and mate fidelity [36].

The egg measurements were conducted during the breeding season of 2011 and involved examining a sample of eggs laid in different nesting habitats and nest types of the Stamfani colony. A Vernier caliper (readable to 0.05 mm) and a digital pocket scale (readable to 0.01 g) were used for the egg measurements, such as weight (WE), length (L), and width (W). The dates of fieldwork in June corresponded to the early stages of laying so as to minimize biases in egg weight measurements. In addition, egg size was also estimated according to the volume index, $L \times W^2$ (length and width in mm), so as to make comparisons with other colonies [37].

### 2.4. Statistical Analysis

One-way ANOVA was applied to each egg variable to evaluate differences according to breeding performance. Prior to the ANOVA, we examined data for assumptions of normality and homogeneity of variance using the Kolmogorov–Smirnov and Levene tests, respectively.

To explore the trends of success per breeding stage, the non-parametric Mann–Kendall test and Sen's slope estimator were applied to detect and estimate the trend magnitude within a time-series [38]. The Mann–Kendall (MK) test is a non-parametric and distribution-free test that indicates only the trend direction (null, increasing, decreasing) over time. Additionally, Sen's slope estimator was used to calculate the magnitude of trends [39]. The unit of Sen's slope is the slope magnitude per year. MK tests were applied with the R programming language [40] and the Kendall and trend packages [41,42].

Abiotic and biotic factors, such as type of nest sites, nest-entrance orientation, and nesting habitat (sectors), reflecting different levels of rat presence and possible predation were also used as independent variables in order to model the breeding performance of the Scopoli's Shearwater in Stamfani during the 2008–2012 period. More specifically, to assess the effect of abiotic and biotic factors on reproductive success, nests were ranked according to their frequency of use by reproductive season. Low-quality nests were defined to have been successfully used for zero or one year, medium-quality nests for two or three years, and high-quality nests for four or five years. Ordered Logistic Regression (OLR) was then used to model the probability of each nest site being in a particular breeding-quality class (low, medium, or high). The model estimates coefficients for each independent variable to determine how it affects the likelihood of being placed in a higher category. The data were partitioned according to the observed covariate patterns using the categorical covariates only.

More specifically, the first stage of the OLR modeling process included the classification of nests belonging to different breeding classes (low, medium, high) per nest type, nest-entrance orientation, and habitat sector during the monitoring of Scopoli's Shearwater breeding performance on Stamfani Island for all breeding seasons (Figure 3). In order

to model the probability of each nest site being in a particular quality class, we used as a reference (baseline) category the group of categorical variables that showed the lower representation for each predictor variable, i.e., "Shrub cover" for nest type, "North" for nest-entrance orientation, and "East" for coastline-habitat sector. In the second stage, we determined the impact of predictor variables, (a) separately, and (b) as a group, on the likelihood of a nest site being placed in a higher nest quality class, which means higher breeding success. Furthermore, we compared the overall performance of all OLR models by using different evaluation metrics, such as Pseudo R-squared (McFadden), Residual Deviance, Akaike Information Criterion (AIC), and the Chi-squared Test.

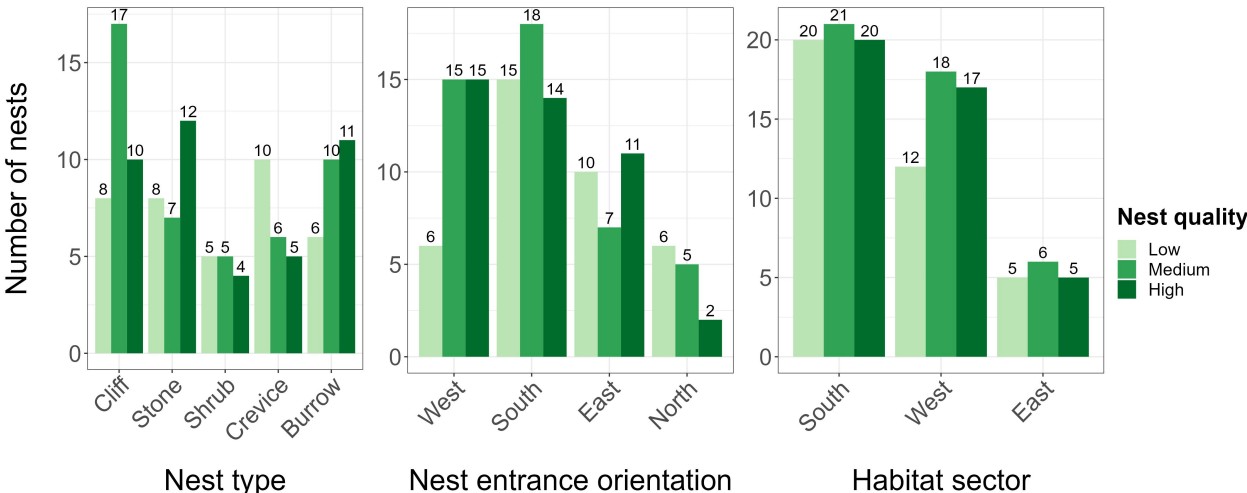

**Figure 3.** Number of nests belonging to different breeding classes per nest type, nest-entrance orientation, and habitat sector during the monitoring of Scopoli's Shearwater breeding performance on Stamfani Island (period: 2008–2012).

The Pulkstenis–Robinson chi-squared test was used as a goodness-of-fit test for the OLR models since it is capable of accommodating models with continuous as well as categorical predictors. The models were developed in R using the polr function of the MASS package [43], DescTools for PseudoR2 [44], and Generalhoslem for goodness of fit [45]. The values of the variables are given as mean ± s.d.

## 3. Results

### 3.1. Breeding Phenology

The Scopoli's Shearwater colony of Stamfani Island showed a high degree of breeding synchrony and nest site tenacity. The birds return to the Strofades islands every year after having spent 4.5 months in the tropical waters off the coast of West Africa and the equatorial waters of the eastern Atlantic. This was based on telemetry data that revealed an almost synchronized departure from the breeding site (24th–25th of October), but a significant spread of departure dates (1st–26th of February) to return from the wintering areas to the vicinity of their colony in the Ionian Sea [46]. The pre-breeding season lasts from late February to mid-May when Scopoli's Shearwaters are recovering from energy losses due to migration demands, socializing during the formation of rafts, mating, and coming ashore for relatively short periods during moonless nights so as to acquire and defend nest sites of high quality. Egg-laying takes place between the 25th of May and the 15th of June, while about 80% of females lay their eggs until the 5th of June (Figure 4). The eggs usually hatched between the 12th and the 31st of July, but the overwhelming majority (90–95%) hatched by the 20th of July. The fledglings are ready to abandon their nests during the last 15 days of October in order to migrate to their wintering areas for the first time (Figure 5).

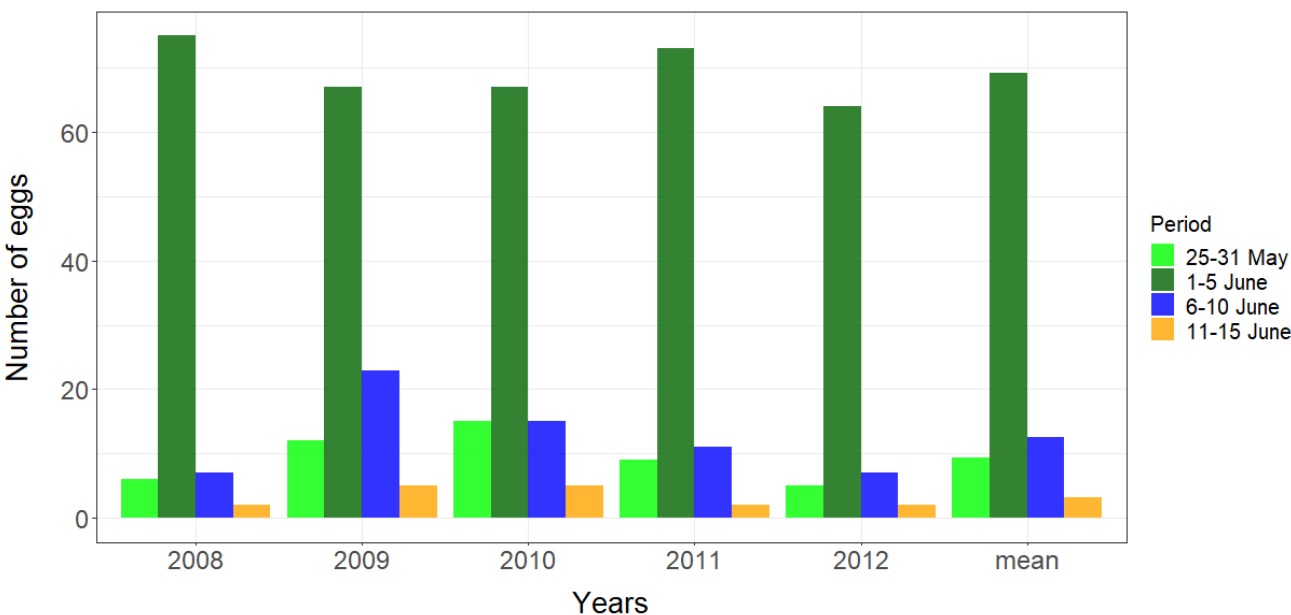

**Figure 4.** Number of Scopoli's Shearwater eggs laid per laying period on Stamfani Island during five consecutive years (2008–2012). Mean scores are also shown.

| Scopoli's Shearwater annual cycle - Stamfani Island | | | | | | | | | | | | | | | | | | | | | | | | |
|---|---|---|---|---|---|---|---|---|---|---|---|---|---|---|---|---|---|---|---|---|---|---|---|---|
| January | | February | | March | | April | | May | | June | | July | | August | | September | | October | | November | | December | |
| 1 - 15 | 15 - 31 | 1 - 15 | 15 - 28 | 1 - 15 | 15 - 31 | 1 - 15 | 15 - 30 | 1 - 15 | 15 - 31 | 1 - 15 | 15 - 30 | 1 - 15 | 15 - 31 | 1 - 15 | 15 - 31 | 1 - 15 | 15 - 30 | 1 - 15 | 15 - 31 | 1 - 15 | 15 - 30 | 1 - 15 | 15 - 31 |

FIGURE LEGEND

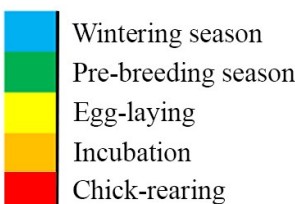

Wintering season
Pre-breeding season
Egg-laying
Incubation
Chick-rearing

**Figure 5.** Scopoli's Shearwater annual cycle including the phases of breeding phenology on Stamfani Island.

### 3.2. Breeding Performance

The data obtained by monitoring 472 nests of Scopoli's Shearwater during five consecutive years (2008–2012) showed a breeding success (fledging per occupied nest per year) of up to 66.60 ± 10.24%. In addition, hatching success (chick hatched successfully per egg laid) was 76.90 ± 4.22%, and fledging success (fledging young per chick hatched successfully) was 86.80 ± 3.39% (Figure 6). An annual variation in breeding performance was also observed during the sampling period. More specifically, the breeding season of 2010 (82.35% breeding success; N = 102 nests) and the breeding season of 2011 (54.74% breeding success; N = 95 nests) represented the best and the worst seasons, respectively, in terms of breeding performance. On the other hand, the breeding seasons of 2008 (N = 90 nests),

2009 (N = 107 nests), and 2012 (N = 78 nests) showed similar scores of breeding success, ranging between 62 and 69% (Table S1). For all breeding seasons, fledging success was higher compared to hatching success.

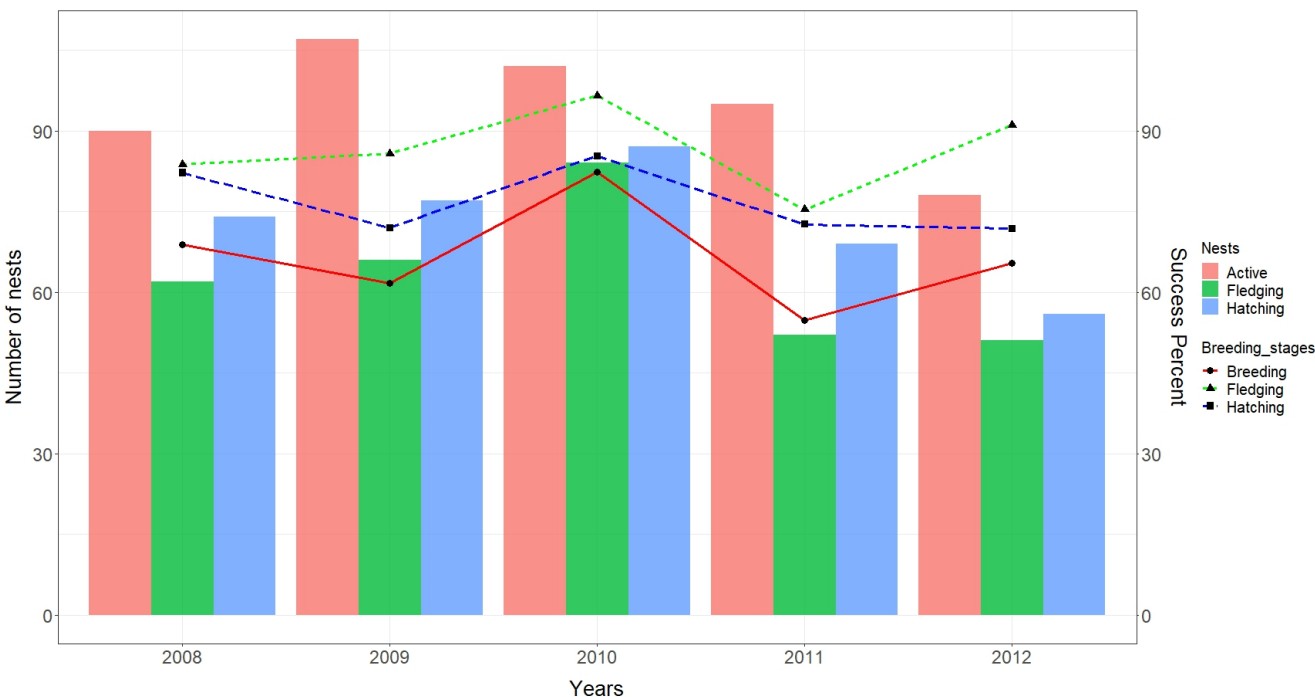

**Figure 6.** Scopoli's Shearwater breeding performance on Stamfani Island. Annual levels of breeding, hatching, and fledging success during five consecutive years (2008–2012) are shown.

The Mann–Kendall (MK) test did not reveal any statistically significant monotonic trends in breeding performance. More specifically, hatching success showed a non-significant negative trend (MK: tau = −0.4, $p$ = 0.46, Sen slope = 1.72). Fledging and breeding success had a more stable trend (MK: tau = 0.2 and tau = −0.2, respectively, $p$ = 0.81, Sen slope = 1.8 for fledging success, and Sen slope = −2.17 for breeding success). Predation pressure by black rats, feral cats (*Felis catus*), and yellow-legged gulls (*Larus michahellis*) have been observed thereon and may be partially responsible for the breeding failure of nests. Nevertheless, the majority (60–70%) of the breeding failures were attributed to the abandonment of nests, mainly occurring during incubation and in the hatching period (Karris per. obs.).

In June 2011, a total sample of 30 eggs laid in different sub-colonies of Stamfani Island were examined and the length, width, and weight scores were estimated at 6.75 ± 0.19 cm, 4.55 ± 0.14 cm, and 76.40 ± 4.69 g, respectively (Table S2). In addition, the egg size, estimated by the volume index $LxW^2$, was 139.9 ± 10.01. The length of the eggs was also found to be smaller in the nests with a higher breeding success (one-way ANOVA; $F$ = 4.400; $p < 0.05$) (Table S3; Figure S1).

### 3.3. Factors Influencing Breeding Success

The results of the OLR models based on nest type, nest-entrance orientation, and habitat sector separately are presented in Tables 1–3 respectively, while the relevant results of the OLR model for the group of all predictor variables are presented in Table 4 (see also Figure S2 for the R code).

**Table 1.** OLR model based on nest type as predictor variable on the likelihood of a nest site being placed in a higher nest quality class. Shrub cover nest was used as reference (baseline) category as the less representative nest type in the sample. CI indicates confidence interval scores.

| Model Breeding Success Classes~Nest Type | | | | | |
|---|---|---|---|---|---|
| | Estimate | Std. Error | t Value | *p*-Value | Odds Ratio (95% CI) |
| Intercept | | | | | |
| Low-quality nests/Medium-quality nests | −0.381 | 0.304 | −1.252 | 0.211 | 0.682 |
| Medium-quality nests/High-quality nests | 0.583 | 0.306 | 1.904 | 0.056 | 1.792 |
| Coefficient | | | | | |
| Cliff cover nests | 0.179 | 0.352 | 0.508 | 0.611 | 1.196 (0.599–2.391) |
| Stone cover nests | 0.316 | 0.371 | 0.852 | 0.393 | 1.372 (0.663–2.844) |
| Burrow nests | 0.363 | 0.369 | 0.983 | 0.325 | 1.438 (0.697–2.973) |
| Crevice nests | −0.245 | 0.389 | −0.628 | 0.529 | 0.782 (0.364–1.679) |

**Table 2.** OLR model based on nest-entrance orientation as predictor variable for the likelihood of a nest site being placed in a higher nest quality class. Northern nest-entrance orientation was used as reference (baseline) category as the less representative nest-entrance orientation in the sample. CI indicates confidence interval scores. * $p < 0.05$, ** $p < 0.01$.

| Model Breeding Success Classes~Nest-Entrance Orientation | | | | | |
|---|---|---|---|---|---|
| | Estimate | Std. Error | t Value | *p*-Value | Odds Ratio (95% CI) |
| Intercept | | | | | |
| Low-quality nests/Medium-quality nests | −0.045 | 0.317 | −0.144 | 0.885 | 0.955 |
| Medium-quality nests/High-quality nests | 0.925 ** | 0.325 | 2.847 | 0.004 | 2.523 |
| Coefficient | | | | | |
| Eastern orientation | 0.491 | 0.381 | 1.289 | 0.197 | 1.635 (0.777–3.471) |
| Southern orientation | 0.409 | 0.354 | 1.155 | 0.248 | 1.506 (0.754–3.038) |
| Western orientation | 0.792 * | 0.367 | 2.153 | 0.031 | 2.208 (1.079–4.569) |

**Table 3.** OLR model based on habitat sector as predictor variable for the likelihood of a nest site being placed in a higher nest quality class. Eastern habitat sector was used as reference (baseline) category as the less representative habitat sector in the sample. CI indicates confidence interval scores.

| Breeding Success Classes~Habitat Sector | | | | | |
|---|---|---|---|---|---|
| | Estimate | Std. Error | t Value | *p*-Value | Odds Ratio (95% CI) |
| Intercept | | | | | |
| Low-quality nests/Medium-quality nests | −0.473 | 0.285 | −1.658 | 0.097 | 0.622 |
| Medium-quality nests/High-quality nests | −0.473 | 0.285 | 1.658 | 0.097 | 1.606 |
| Coefficient | | | | | |
| Western sector | 0.148 | 0.323 | 0.460 | 0.645 | 1.160 (0.615–2.187) |
| Southern sector | 0 | 0.313 | 0 | 0.999 | 0.999 (0.540–1.851) |

**Table 4.** OLR model based on all predictor variables for the likelihood of a Scopoli's Shearwater nest site being placed in a higher nest quality class. "Shrub cover" for nest type, "North" for nest-entrance orientation, and "East" for coast-line-habitat sector eastern habitat sector were the group of categorical variables used as the mixed reference (baseline) category as the less representative in the sample. CI indicates confidence interval scores. * $p < 0.05$.

| Model Breeding Success Classes~Nest Type + Nest Orientation + Habitat Sector | | | | | |
|---|---|---|---|---|---|
| | Estimate | Std. Error | t Value | *p*-Value | Odds Ratio (95% CI) |
| Intercept | | | | | |
| Low-quality nests/Medium-quality nests | 0.167 | 0.548 | 0.305 | 0.759 | 1.182 |
| Medium-quality nests/High-quality nests | 1.157 * | 0.554 | 2.087 | 0.036 | 3.183 |
| Coefficient | | | | | |
| Cliff cover nests | 0.144 | 0.363 | 0.398 | 0.690 | 1.155 (0.567–2.358) |
| Stone cover nests | 0.285 | 0.391 | 0.731 | 0.464 | 1.331 (0.619–2.865) |
| Burrow nests | 0.316 | 0.394 | 0.802 | 0.422 | 1.372 (0.633–2.981) |
| Crevice nests | −0.272 | 0.399 | −0.683 | 0.494 | 0.761 (0.347–1.664) |
| Eastern orientation | 0.537 | 0.396 | 1.355 | 0.175 | 1.711 (0.789–3.742) |
| Southern orientation | 0.382 | 0.366 | 1.043 | 0.296 | 1.465 (0.717–3.017) |
| Western orientation | 0.692 | 0.389 | 1.777 | 0.075 | 1.998 (0.935–4.314) |
| Western Sector | 0.168 | 0.404 | 0.417 | 0.676 | 1.184 (0.536–2.626) |
| Southern Sector | 0.127 | 0.346 | 0.367 | 0.713 | 1.135 (0.576–2.246) |

The comparison of the models showed that none of the OLR models provides a significant improvement in fit compared to the null model so as to make it a preferable choice for explaining the variation in breeding success classes based on the metrics given in Table 5. Moreover, the model for the group of all predictor variables revealed the highest McFadden score (0.038) and the lowest Residual Deviance (262.739), indicating a better fit among the models. On the other hand, the OLR model based on nest-entrance orientation showed the lowest AIC score (276.383) among the models, indicating the best trade-off between the complexity of the model against how well the model fits the data.

**Table 5.** Overall performance of OLR probability models of each Scopoli's Shearwater nest site being in a particular quality class based on different evaluation metrics, including Pseudo R-squared (McFadden), Residual Deviance, Akaike Information Criterion (AIC), and Chi-squared Test.

| Models | PseudoR$^2$ | | | Residual Deviance | AIC | Goodness of Fit | | |
|---|---|---|---|---|---|---|---|---|
| | McFadden | CoxSnell | Nagelkerke | | | Chi-Squared | df | *p*-Value |
| Breeding success classes~Nest type | 0.015 | 0.034 | 0.038 | 267.323 | 279.323 | 3.764 | 6 | 0.708 |
| Breeding success classes~Nest-entrance orientation | 0.019 | 0.041 | 0.046 | 266.383 | 276.383 | 2.654 | 4 | 0.617 |
| Breeding success classes~Habitat sector | 0.002 | 0.004 | 0.005 | 271,141 | 279.141 | 0.217 | 2 | 0.897 |
| Breeding success classes~A | 0.038 | 0.069 | 0.078 | 262.739 | 284.739 | 74.653 | 74 | 0.456 |
| A = Nest type + Nest-entrance orientation + Habitat sector | | | | | | | | |

Focused on the OLR model for the group of all predictor variables, the odds ratio of 1.182 for "Low-quality nests/Medium-quality nests" indicates that, compared to the

reference category, the odds of moving to a higher breeding success class are about 18.2% (Table 4). Furthermore, the odds ratio of 3.183 for "Medium-quality nests/High-quality nests" indicates that, compared to the reference category, the odds of moving to a higher breeding success class are about 218.3% (t value = 2.087; $p < 0.05$). This suggests a significant increase in the odds of higher breeding success for nests classified as "Medium-quality nests/High-quality nests" compared to the lower classes. It is also worth mentioning that (a) the odds ratio for burrow nests suggests a greater increase (37.2%) in the odds of greater breeding success for nests of this type compared to the reference category "Shrub cover", (b) the odds ratio for the western habitat sector suggests a greater increase (18.4%) in the odds of greater breeding success for nests of that sector compared to the reference category "East" for the coastline-habitat sector, and (c) the odds ratio for the western nest-entrance orientation suggests a greater increase (99.8%) in the odds of greater breeding success for nests with a western nest entrance compared to the reference category "North". The latter is in accordance with the outcome of the OLR model based on nest-entrance orientation, where the odds ratio for the western nest-entrance orientation suggests a significant increase ($p < 0.05$) in the odds of greater breeding success for nests compared to the reference category "North". In conclusion, the overall outcome shows that breeding success was influenced by the type of nest site and the nest-entrance orientation, whereas the nesting habitat did not influence breeding success.

## 4. Discussion

### 4.1. Breeding Phenology

The first systematic monitoring of the reproductive effort of Scopoli's Shearwater on Stamfani Island provided baseline information on the breeding cycle of the species in that area, an area that hosts the largest species colony in Greece [27]. The crucial dates of breeding stages found here are closely aligned with corresponding studies carried out in other Mediterranean colonies of the species [31,33,47,48]. Incubating was biparental, as was also confirmed by geolocators deployed on breeders, where parents usually alternate the incubation role every eight to nine days during June [46]. These results are in accordance with relevant findings regarding the related species Cory's Shearwater (*Calonectris borealis*) in the Atlantic [10], highlighting in this way the importance of continuous egg incubation but also the need to replenish the inevitable energy losses of breeders. The duties of both parents are still particularly increased during the first stages of chick rearing and especially in the first two weeks after the eggs hatch since almost every night the parents visit their nests for food provision [16]. These visits gradually thin out, with the absence of the parents being noticeable during the last stages of the fledging period and especially after the 20th of September.

### 4.2. Breeding Performance

The average breeding success, for five consecutive breeding seasons (2008–2012), was found to be about 66–67%. This outcome is comparable with respective results from other Scopoli's Shearwater colonies, such as those on the Dionysades islands north of eastern Crete (77%), the Marseille islands (79–82%), the Maltese islands (64%), and Linosa Island, close to Sicily (39–89%) [33,49–51]. Predation pressure by black rats was the main reason for the low level of breeding success (39%) on Linosa Island just before a rat eradication project was launched, but this was not the case for the breeding failures on Stamfani Island. The by-catch incidental mortality of Scopoli's Shearwater seems to be a significant threat for the species in the southern Ionian Sea, and a possible reason for the sudden end to breeding effort, since about 1.7–2.0% of the Strofades population has been estimated to have been caught in longline fishery operations [19]. As a result, the accidental trapping of shearwaters could be considered a potential risk for the conservation of the Strofades population by taking into consideration the ecological features of the study species, such as the long-term mate fidelity, the biparental care during the incubation, and the consequent chick-rearing duties.

The low number of abandoned nests after the hatching period, and the early stages of chick rearing, could also be attributed to inexperienced breeders which also tend to change nests. It is worthwhile to mention that the highest level of breeding success (77.5%) was found in the group of nests that were occupied throughout the five breeding seasons (N = 48; 30.38% of the total number of monitored nests; Figure S3). These nests were mainly occupied by experienced males, according to our ringing recoveries, which could explain such a high level of breeding success [11,36,52]. Indeed, a recent study has shown that experienced males rarely change nests, and seldom do so in response to the results of previous breeding performances [53].

The shearwaters' breeding performance showed interannual variations, but no significant monotonic trends, as already described. It is assumed that this outcome was obtained due to the short time series of the data. Nevertheless, it did provide valuable information on success trends (positive or negative, depending on the sign of the Sen slope value). The collection of more data in subsequent breeding seasons, as well as the further study of nest site tenacity, can provide explanations for the variation in breeding success on Stamfani Island.

The mean length, mean width, and mean size index of the Scopoli's Shearwater eggs on Stamfani Island were found to be similar to those from another Greek colony located on a satellite islet of Crete [54]. The aforementioned results also enhance relevant findings from other studies that support biometric clinal variation in Scopoli's Shearwater from eastern to western Mediterranean colonies [55–58]. That morphological pattern is in accordance with the eastward decline in primary production and increase in sea surface temperature of the Mediterranean Sea [59]. It is assumed that philopatric behavior, mate fidelity, and nest site tenacity, as well as discrete distribution of the remote insular breeding areas, may lead to the maintenance of morphometric variation among colonies. Additionally, these morphological traits of the species could be related to the spatiotemporal fluctuations of ecogeographical factors that shape the conditions of foraging areas used by different colonies spread around the Mediterranean.

In the current study, breeding success was negatively linked to egg length. This outcome is not in line with relevant findings of Cory's Shearwater in Selvagem Grande where big eggs had higher possibilities of successful hatching and successful reproductive output than smaller ones [37]. Even if the least experienced females tend to lay smaller eggs and improve their ability to produce larger eggs with age [60–62], we assumed that elongated eggs could be more susceptible to damage during incubation. As a consequence, we propose further research into the influence of egg size, the experience of the breeders, and incubating behavior on breeding success.

### 4.3. Factors Influencing Breeding Success

Breeding habitat, nest type, and nest-entrance orientation are among the factors that may affect nesting in terms of heat retention, humidity insulation, and light penetration, as well as access for predators and exposure to atmospheric forces, such as strong winds and rains, and as a result, may have an impact on the breeding output of burrow-nesting seabirds [52,63,64]. It is also evident that the cavity selection of good-quality nest sites is a significant predictor of breeding success in Procellariiformes [65,66]. Our results show that shearwaters prefer to use cliff-cover nests, stone-cover nests, and burrow nests, equally classified as medium-quality (38%) and high-quality (37%) nest sites (Figure 3). These nests have as a common feature the existence of soft soil and thus a substrate for excavation that allows for greater protection of eggs and chicks by reducing the possibility of detection by potential predators [67,68]. The higher occupancy rates for the above-mentioned nest types may possibly be biased by the lower detection rate for breeders using shrub-cover nests and crevices. Nevertheless, we believe that the detectability of large, easily accessible burrow-nesting species such as Scopoli's Shearwater is high and not biased by the nest type.

Regarding the impact of nest-entrance orientation, we found that a western nest-entrance orientation suggests a significant increase in the odds of greater breeding success

for nests compared to the reference category "North". As was already described in the Results section, a northern nest-entrance orientation was used as the reference category as it was the less representative orientation in the sample. Indeed, shearwaters seemed to avoid nests with northern nest entrances, perhaps as a strategy to avoid exposure to prevailing northern cold winds. We also assumed that this outcome could reflect the need of birds to facilitate landing and take-off processes through drag force and lift force, respectively, thus selecting nesting sites with entrances opposite to the sea [67].

We found no significant impact of the coastline-habitat sector on Scopoli's Shearwater breeding success. This result reflects an imperceptible footprint of rat presence, and consequently predation, on reproductive outcomes. Black rats, which are categorized as omnivorous but most of the time are herbivorous, have a short reproduction period where breeding mainly occurs in May and June [29], thus having the highest population density in July-September. Therefore, the period of possible egg and chick predation coincides with a decrease in vegetation productivity and relevant biomass due to summer drought [23]. Nevertheless, the study area shows high levels of humidity in summer which significantly reduce vegetation dryness and maintain its density. As a result, the black rat may continue to act as a herbivorous species even during the summer months. Moreover, the study area constitutes a limestone island that offers a high availability of suitable nesting sites (cavities and burrows) while the smoother texture of limestone rock on the coastline results in a large number of sites protected from rat predation [69]. We assume that this could be a possible explanation for the non-significant effect of rat predation on Scopoli's Shearwater in the study area, which does not constitute a typical Mediterranean dry insular site.

Overall, the current study provides the first data on the breeding biology and success rate for Scopoli's Shearwater in the Strofades island group, which constitutes one of the largest colonies in European territory. Effective conservation measurements and assessments at global and local scales need such fundamental knowledge, which can be used as baseline data for the evaluation of possible impacts by forthcoming activities related to hydrocarbon exploration and the construction of offshore wind farms in the Ionian Sea. Therefore, a long-term research effort is needed in order to examine more factors as possible determinants of breeding success and to identify relevant trends in more detail.

**Supplementary Materials:** The following supporting information can be downloaded at: https://www.mdpi.com/article/10.3390/d16030150/s1, Figure S1: Box plot showing median, interquartile range, and range referring to the egg length of Scopoli's Shearwater nest sites revealed breeding failure and breeding success (breeding season 2011, N = 30 nests); Figure S2: R code developed for Ordered Logistic Regression (OLR) analysis used to model the probability of each Scopoli's Shearwater nest site being in a particular quality class (low, medium, or high); Figure S3: Classification of the Scopoli's Shearwater nests that were monitored on Stamfani Island per occupation rate during 2008–2012; Table S1: Breeding performance of Scopoli's Shearwater on Stamfani colony during different reproductive stages (sampling period: 2008–2012); Table S2: Egg measurements of Scopoli's Shearwater colony on Stamfani Island (sampling period: 2–5 June 2011); Table S3: Com-parison of egg measurements and egg volume index between nest sites with different breeding performance (suc-cess-failure) on Stamfani Island (one-way ANOVA). * $p < 0.05$

**Author Contributions:** Conceptualization, G.K.; methodology, G.K., S.X. and KP; software, G.K., K.P. and A.T.; validation, G.K., S.S. and S.X.; formal analysis, G.K. and K.P; investigation, G.K., S.X. and M.-D.V.; resources, G.K. and S.X.; data curation, G.K., S.X. and K.P.; writing—original draft preparation, G.K., K.P. and A.T.; writing—review and editing, G.K., S.X., K.P., S.S., S.G. and M.-D.V.; visualization, G.K., K.P. and A.T.; supervision, S.S. and S.G.; project administration, G.K.; funding acquisition, G.K and S.X. All authors have read and agreed to the published version of the manuscript.

**Funding:** This study was partially funded by LIFE07/NAT/GR/000285 and the A.G. Leventis Foundation.

**Institutional Review Board Statement:** The current research expeditions had specific permit every year (breeding season) from the Management Agency of the National Marine Park of Zakynthos (NMPZ), which as public service belongs to the Greek Ministry of Environ-ment and Energy (e.g. ref. no. 667/14.05.2010/NMPZ; 517/10.04.2012/NMPZ). Additionally, the Natural History Mu-seum of Crete (scientific institution code GR002), partner of the current research study, has a CITES (the Convention on International Trade in Endangered Species of Wild Fauna and Flora) sampling permit for wildlife (ref. no. 096860/2199/23-8-2005). All the relevant activities-samplings were carried out ensuring the maximum safety of Sco-poli's Shearwater as well as the least possible disturbance to adults and fledglings of the target species and its breeding habitat. Furthermore, it was our obligation to compile a relevant detailed technical report of the samplings and results that we achieved in an annual base so as to send it to the Management Agency of the NMPZ.

**Data Availability Statement:** The data supporting the results or analyses presented in the paper can be found in the Department of Environment, Ionian University (contact person: Georgios Karris).

**Acknowledgments:** We would like to thank the local authorities, e.g., the Management Agency of the National Marine Park of Zakynthos, the National Coastguard Authorities, and the Metropolis of Zakynthos, for giving permission and providing assistance to study the Scopoli's Shearwater on the Strofades islands, and the Hellenic Bird Ringing Centre for providing us with rings. In addition, we would like to thank Kostas Grivas, Dimitrios Tziertzidis, Martin Plappert, and Kostas Gaganis for their contribution to fieldwork.

**Conflicts of Interest:** The authors declare no conflicts of interest.

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
