# Peer review of "Aspects of Breeding Performance of Scopoli’s Shearwater (Calonectris diomedea): The Case of the Largest Colony in Greece"

_diversity, doi:10.3390/d16030150_

Round 1

Reviewer 1 Report

Comments and Suggestions for Authors

Author Response

Cover letter with revisions diversity-2875653

“Aspects of breeding performance of Scopoli’s Shearwater (Calonectris diomedea): The case of the largest colony in Greece”

Dear Reviewer,

On behalf of all co-authors of the submitted ms diversity-2875653, I send you below detailed replies (red text) to your comments. I also attach the revised manuscript with all revisions highlighted. 

Yours sincerely,

G. Karris

Review 1

Curious that the paper was submitted to this journal rather than an ornithological or zoological one. Nevertheless, I found the Results exquisite and well presented! Lns 263-375.

The Intro and Discussion, though, are over-written and could well be seriously shortened. A number of extraneous issues are reviewed. I’m for just getting to the goods, so to speak. Otherwise, nice job on the study and the Results. Good stuff.

Thank you very much for you useful comments and suggestions. You may check the revised text and our relevant clarifications-replies below.

Ln 17-19, 38-41. I review quite a lot of seabird papers and these days a huge proportion start out this way. Can’t we just deal with the species and its natural history, which the paper contributes to immensely, and forget about trying to somehow make the study even more, but indirectly relevant? No need to feel guilty about increasing the understanding of a seabirds’ natural history patterns, just for the mere sake of increasing knowledge.

Maybe begin with last 4 words on ln 43, that sentence more or less repeating the preceding one, and just say that k-selection allows it to cope with varying conditions (including occasional breeding failure).

We erased Lines 17-19 as you suggested. On the other hand we believe that is necessary for the international audience to mention that seabirds are nowadays recognized as one of the most threatened group of birds and conservation measures based on their breeding ecology constitutes an urgent need.

Ln 47-48. Mentioning the albatross seems somewhat tangential.

We agree, we erased that part for albatross.

Ln 51-53. This is said repeatedly but actually seabirds are rather poor bio-indicators because in most cases very little is known of their diet and its annual variation, especially those species that frequent pelagic waters. Therefore, what is it that they indicate?

Thank for your comment. Nevertheless, we don’t agree with your statement. The use of seabirds as bioindicators of marine ecosystems is widely recognized (please check relevant references added in the ms). More specifically they are indicators of fishing grounds and fish stocks as well as means of biomass transport according to their foraging and migration pattern respectively. For example, both physical oceanographic incidents and biotic activity are responsible for seabird prey concentrations over a variety of spatial and temporal scales while numerous studies have examined factors responsible for concentrating aggregations of prey and consequently of seabirds e.g. in the Atlantic Ocean and the Mediterranean Sea so as to determine whether these aggregations are random or predictable. Additionally, It is expected that extreme low values of primary productivity and consequently of food around seabird colonies will force breeders to perform a high proportion of long trips in their effort to reach the most profitable areas for food provision to chicks but also for their own energy demands. As a result, we nowadays pay attention not only to studies for diet analysis of seabirds but also to the use of marine waters base on innovative telemetry tools. 

Ln 54-56. Delete

Done

Ln 76. Begin Intro here and include a sentence revealing this species k-selected attributes. Or you could start with ln 87 and embellish a little. Really, readers will be seabird enthusiasts, mostly, and they know all of how you begin the paper.

We erased Lines 54-75.

Ln 377-384. Delete; readers already know this stuff.

Done

Ln 436-445. Delete, superfluous and just conjecture not related to actual findings of this study. So what, lots of seabirds respond to ENSO/NAO. It’s actually resulting variation in food availability to which they mostly are responding, and we know very little about that for most seabirds. Your results seem to infer that nest predation plays a major role in nesting success. Leave it at that for now, until you have maybe 20 years of data.

Done, we erased the relevant text.

Reviewer 2 Report

Comments and Suggestions for Authors

The major problem with this manuscript in misplacing certain parts of text. for example: the species presentation should be in 'Introduction' not in 'Methods' section. Data on factors affecting breeding succees (predation etc.) should be in 'Results', not in 'Discussion' I have indicated these misplacemnets directly in the text.

The other problem is concerning factors influencing breeding success, direct factors (predators, weather conditions) seems to be here of less importance than indirect factors, such as nesting sites characteristics. These should be clearly separated. 

There is an inconsistency in writting common species names (with capital letter or small, different rules for bird names, differnt for mammals, etc.).  Definitions: breeding success, hatching success, fleding success, nesting success, reproductive success, should be cleraly explaind in 'Methods' section. In addition to these parameters, other indictaros should be also calculated (overall and for each year): the mean number of hatchlings (and separately of fledglings) per breeding pair (and separatley for successful pair). Also detailed phenology of egg-laying period, overall and for in each year (preferable in each pentads, like 20-25 May, 26-30 May, 1-5 June, etc..., with the number of clutches in each of these pentades) should be illustrated. It should also be emphesized in 'Results' that all pairs (or 95%??) laid one egg only.       

Author Response

Cover letter with revisions diversity-2875653

“Aspects of breeding performance of Scopoli’s Shearwater (Calonectris diomedea): The case of the largest colony in Greece”

On behalf of all co-authors of the submitted ms diversity-2875653, I send you below detailed replies (red text) to your comments. I also attached the revised manuscript with all revisions highlighted. 

Yours sincerely,

G. Karris

Review 2

The major problem with this manuscript in misplacing certain parts of text. for example: the species presentation should be in 'Introduction' not in 'Methods' section.

Data on factors affecting breeding success (predation etc.) should be in 'Results', not in 'Discussion' I have indicated these misplacements directly in the text.

Thank you very much for you useful comments and suggestions. You may check the revised text and our relevant clarifications-replies below. We have changed the structure of the ms according to your suggestions. We have also erased the lines 54-75 which you suggested to be significantly shortened.

The other problem is concerning factors influencing breeding success, direct factors (predators, weather conditions) seems to be here of less importance than indirect factors, such as nesting sites characteristics. These should be clearly separated.

Done. We have added to Results sub section “3.3. Factors influencing breeding success”, that:  “In conclusion, the overall outcome shows that breeding success is influenced by the type of the nest site and the nest entrance orientation, whereas the nesting habitat did not.”

There is an inconsistency in writing common species names (with capital letter or small, different rules for bird names, different for mammals, etc.). 

Done, we have corrected by using capital letter to all common species names and italics to latin names.

Definitions: breeding success, hatching success, fledging success, nesting success, reproductive success, should be clearly explained in 'Methods' section.

Definitions are given to Sub-section 2.2 “Breeding performance”. More specifically we clarify that “According to the obtained data, breeding success (% fledglings per egg laid), hatching success (% chicks hatching successfully per egg laid) and fledging success (% fledglings per egg laid) were estimated.”

In addition to these parameters, other indictors should be also calculated (overall and for each year): the mean number of hatchlings (and separately of fledglings) per breeding pair (and separately for successful pair).

Done, please check Table S1 in Supplementary material.

Also detailed phenology of egg-laying period, overall and for in each year (preferable in each pentads, like 20-25 May, 26-30 May, 1-5 June, etc..., with the number of clutches in each of these pentades) should be illustrated.

Done. Please check the new Figure 4 added to the ms.

It should also be emphasized in 'Results' that all pairs (or 95%??) laid one egg only.      

Scopoli’s Shearwater lay only one egg per pair and this information was added in the Introduction section.

My general comment to this section (Results section) is: too much focus on statistics as such, and not enough attention is paid to real factors shaping the breeding success. It look as if you do not have such data, but if such details are available you should tabulate them and focus on these rather than on the 'associates'.

The current study focused on the possible relationship between breeding success and specific environmental factors, namely type of nest sites, nest-entrance and orientation features, nesting habitat reflecting different level of rat presence and possible predation, rate of nest site occupation throughout sampling years and its correlation with breeders’ experience, and dimensions of eggs.  As we have declared, the current study provides the first data on the breeding biology and success rate for Scopoli’s Shearwater on Strofades island group and that collection of more data in subsequent breeding seasons e.g. further study of nest site tenacity, can provide further explanations for the variation of breeding success on Stamfani Island.

Reviewer 3 Report

Comments and Suggestions for Authors

row 111. A few sentences about the human presence past and present on these islands would also be advisable. The current state of the buildings on the island is also worth describing, as it may have implications for future activities on the site.

Figure 1. I would expect to see photographs of the habitats in each sector. Then there will be full documentation of these surveys.

Figure 2. Something is wrong with the display of the photos.

Author Response

Cover letter with revisions diversity-2875653

“Aspects of breeding performance of Scopoli’s Shearwater (Calonectris diomedea): The case of the largest colony in Greece”

On behalf of all co-authors of the submitted ms diversity-2875653, I send you below detailed replies (red text) to your comments. I also attached the revised manuscript with all revisions highlighted.

Yours sincerely,

G. Karris

Review 3

row 111. A few sentences about the human presence past and present on these islands would also be advisable. The current state of the buildings on the island is also worth describing, as it may have implications for future activities on the site.

Thank you for your comment. The current study focused on the possible relationship between breeding success and specific environmental factors, namely type of nest sites, nest-entrance and orientation features, nesting habitat reflecting different level of rat presence and possible predation, rate of nest site occupation throughout sampling years and its correlation with breeders’ experience, and dimensions of eggs.  As we have declared, the current study provides the first data on the breeding biology and success rate for Scopoli’s Shearwater on Strofades island group and that collection of more data in subsequent breeding seasons e.g. further study of nest site tenacity can provide further explanations for the variation of breeding success on Stamfani Island. The human presence on the island is limited (the last habitant who was a monk was died few years ago) but we agree that human presence has to be included in the future studies for the conservation of the remote Strofades island group as an ecosystem.

Figure 1. I would expect to see photographs of the habitats in each sector. Then there will be full documentation of these surveys.

We believe that the detailed map of the habitats provides such information about the updated land cover of the study area.  

Figure 2. Something is wrong with the display of the photos

Thank you for your comment. It seems that something went wrong with the convert process from word file to pdf file. Please check the attached word file of the revised ms where the display of Figure 2 is correct.

Round 2

Reviewer 1 Report

Comments and Suggestions for Authors

Great paper, well done and well written!